# Quantification of 21 sugars in tropospheric particulate matter by ultra-high-performance liquid chromatography tandem mass spectrometry

Pauline Bros<sup>1</sup>, Sophie Darfeuil<sup>1</sup>, Véronique Jacob<sup>1</sup>, Rhabira Elazzouzi<sup>1</sup>, Dielleza Tusha<sup>1</sup>, Tristan Rousseau<sup>1</sup>, Julian Weng<sup>2</sup>, Patrik Winiger<sup>2</sup>, Imad El Haddad<sup>2</sup>, Christoph Hueglin<sup>3</sup>, Gaëlle Uzu<sup>1</sup>, and Jean-Luc Jaffrezo<sup>1</sup>

<sup>1</sup>Univ. Grenoble Alpes, CNRS, INRAE, IRD, Grenoble INP, IGE, 38000 Grenoble, France

<sup>3</sup>Empa, Swiss Federal Laboratories for Materials Science and Technology, Laboratory for Air Pollution/Environmental Technology, 8600 Dübendorf, Switzerland

Correspondence to: Sophie Darfeuil (sophie.darfeuil@univ-grenoble-alpes.fr)

**Abstract.** Sugars compose an important class of compounds by mass in atmospheric particulate matter (PM), often with biogenic and anthropogenic sources, many of them still poorly characterized. These sugars are mainly analysed by gas-chromatography coupled to mass spectrometry (GC-MS) or ion chromatography coupled with pulsed amperometric detection (IC-PAD). However, these techniques present several disadvantages such as a complex preparation for GC-MS, or a limited range of possible analytes and elevated limits of quantification for IC-PAD. This hinders our capability to perform analyses of extensive time series, in order to develop our knowledge of the phenomenology of these species. In this paper, we present the validation of an ultra-highperformance liquid chromatography tandem mass spectrometry (UHPLC-MS/MS) method for the simultaneous quantification of 21 sugars in atmospheric PM. The sample preparation is simple, fast and safe, consisting of an aqueous extraction. The method was validated in terms of linearity, accuracy, repeatability, reproducibility, and recovery. This technique showed excellent linearity (r<sup>2</sup>>0.99), precision (relative standard deviation RSD<25%) and extraction yields (results between 70 and 120%). The suitability of the method for analyses of samples from sites with very low PM concentrations was demonstrated with samples from the High-Altitude Research Station Jungfraujoch (JFJ), Switzerland. A series of samples spanning a 6-year period is presented. Results for arabitol, levoglucosan and 2-methyl-tetrols display strong seasonality, due to seasonal variation in chemical production and boundary layer dynamics, with atmospheric convection and transport from the valleys to high altitudes mostly in summer. This simple and fast method facilitates the analysis of large sets of PM samples and sugar compounds, and opens the door to a better understanding and attribution of their sources.

## 1 Introduction







Particulate matter (PM) is involved in many aspects of Human environments, including large impacts on climate change, health, dispersion of pollutants, or damages to all compartments of the critical zone. Some impacts can be positive (the fertilization of Oceans), but most are negative, particularly when it comes to anthropogenic emissions. Assessing these impacts requires a proper understanding of some properties of PM (mass, size distribution, chemistry, physical and optical properties, etc.), and also a good parametrization of the fluxes in and out of the atmosphere, in order to develop Chemical Transport Models (CTM). CTM are a prerequisite for the global assessment of the impacts of the PM in Air Quality, Human health, or climate. However, the main component by

<sup>&</sup>lt;sup>2</sup>Paul Scherrer Institute, PSI Center for Energy and Environmental Sciences, 5232 Villigen, Switzerland

mass of PM, the organic matter (OM), is still poorly understood, even after decades of research on its chemical composition, emissions and transformation processes, and observations worldwide of its variability. CTM are generally off by up to an order of magnitude, compared to measurements (Ciarelli et al., 2016), meaning that sources and/or processes of OM components are not fully captured. Indeed, the current state of the art concerning molecular characterization of OM cannot allow a better mass apportionment than about 25 to 50% at best of the total OM mass (Michoud et al., 2021; Wang et al., 2020). Obviously, several OM sources or formation processes *in situ* are still not understood nor their emission fluxes properly quantified.









While the formation of Secondary Organic Aerosol (SOA) is largely studied by a score of papers every year, the Primary Biogenic Organic Aerosol (PBOA) is largely disregarded, while studies point out its large contribution to the OM fraction (up to 25-35 % of the total OM mass on seasonal average for France) (Samaké et al., 2019). As opposed to Secondary Biogenic Organic Aerosol (SBOA), that is formed in the atmosphere from the biogenic volatile organic compounds (VOCs) emitted by biota, PBOA is a subset of organic PM that comprises all particulate material of biogenic origin directly entering the atmosphere. PBOA includes components like plant debris, fungi, pollen, and viruses (Amato et al., 2017; Elbert et al., 2006; Hummel et al., 2015). The current estimate of the global PBOA flux to the atmosphere is between 10 and 310 Tg per year (Hummel et al., 2015), compared to values of 21 (MACCCity) or 30 (CMIP6) Tg per year for particulate organic components from anthropogenic origins (Rémy et al., 2019). The wide uncertainty on the flux of PBOA indicates by itself the extent of our poor knowledge on this fraction. Moreover, PBOA is currently very rarely included in CTM's (Rémy et al., 2019). One attempt to model PBOA on the European scale using the COSMO-ART CTM (Hummel et al., 2015) showed that PM mass associated with solely fungal spore emissions could represent on yearly average 15% of the total PM (about 25 - 30 % of organic PM) mass over Europe. A couple of papers very recently described the modelling of PBOA from fungal spore emission in CHIMERE (Vida et al., 2024) and EMEP models (Felix-Lange et al., in progress). A better assessment of this PBOA fraction is all the more important, since they also have a significant impact on the oxidative potential of PM, which can be a prevalent source in OP contribution at spring (Samake et al., 2017; Dominutti et al., 2023).

A large part of the PBOA fraction is related to the fungi emission sources (Hummel et al., 2015; Marynowski et al., 2020; Wang et al., 2020). Therefore, the seasonal cycles of some sugars and sugar alcohols (S and SA) concentrations observed in atmospheric PM can be directly related to the microbiology of large biomes. However, this fungi-sugar link has only been documented for a limited number of sugars (e.g. arabitol, mannitol) and cases (Marynowski et al., 2020; Samaké et al., 2020), and the main sources and drivers of many other S and SA in PM have not been identified yet. Currently, the mechanisms of S and SA emissions (along with their full organic cortege) to the atmosphere, and their fate and transfer to other ecosystem compartments remain poorly understood. Anhydrosugars, another form of sugars in atmospheric PM can be natural but also anthropogenic, such as levoglucosan and its isomers mannosan and galactosan, which are commonly used as a tracer for the assessment of PM from biomass burning (natural wildfires, domestic wood heating or cooking). These anhydrosugars – carbohydrate derivatives formed by the elimination of a water molecule - have been of primary importance for source apportionment studies since at least 15 years (Herich et al., 2014; Weber et al., 2019), and are now a basic measurement for PM analysis. The main suspected sources of other sugars measured in atmospheric PM are summarized in Table S1. Many of the links between chemical species and sources proposed in the literature Table S1 are still tentative, with only few comprehensive measurements for many of the species cited.

Analyses of sugars are mostly performed by gas chromatography coupled with mass spectrometry (GC-MS) (Vincenti et al., 2022). This method, with high selectivity and specificity, is generally performed with large sample volume and a sample preparation using derivatization of the hydroxyl groups (Graham et al., 2002). The introduction of ion chromatography coupled with pulsed amperometric detection (IC-PAD) in the mid 2000's (Engling et al., 2006; Yttri et al., 2007) made the determination of sugars in PM easier, at the expense of a narrowing of the range of the accessible sugars due to potential interferences with lack of column separation. Nevertheless, IC-PAD allows for the determination of 8 - 12 S and SA without too many interferences, including the three anhydrosugars (AS) (levoglucosan, mannosan, galactosan) (e.g., Oduber et al., 2021). In recent years, liquid chromatography tandem mass spectrometry (LC-MS/MS) has become more common in laboratories due to its greater efficiency and resolution, speed, simplicity, and low cost compared to more conventional chromatographic techniques (López-Ruiz et al., 2019). Indeed, LC-MS/MS can offer a much larger chemical speciation of S and SA and much lower limits of quantification than IC-PAD, together with much simpler sample preparation than GC-MS.









The analysis of sugars by LC-MS/MS is not a new topic, as they have been quantified for several years in different fields including food, plant and forensic fields. Indeed, fructose, glucose and sucrose are widely analysed in plant and food using such methods: in plants such as arabidopsis and rice (Ito et al., 2014), potatoes and strawberry (Georgelis et al., 2018), grapes (Gika et al., 2012), in bread (Nielsen et al., 2006), fruit juices (Rego et al., 2018), honeydew (Nguyen et al., 2020) and fermentation processes (Pismennõi et al., 2021). In a completely different field, a LC-MS/MS method was validated to analyse eleven sugars and sugar alcohols related to explosive (Tsai et al., 2022). The most comprehensive methods reported to date allow the simultaneous analysis of no more than 19 sugar compounds (Gika et al., 2012; Ito et al., 2014). Even if application areas were numerous, the implemented LC-MS methods were essentially the same. Indeed, the LC separation was performed in an HILIC (Hydrophilic Interaction Liquid Chromatography) mode with column coated with amine or polyamide groups in order to enhance the polar compound retention and selectivity. Moreover, the MS was operated in negative mode and mostly using multiple reaction monitoring (MRM). In summary, these runs exhibit certain similarities with our intended approach, but remain constrained with respect to the range of analytes quantified and are most likely restricted to substantially higher concentration levels than expected for atmospheric PM samples, though probably impacted by more complex matrices. These observations collectively indicate that our experimental design is feasible; however, its implementation require substantial methodological refinement and further development to extend the analytical scope, particularly to encompass a broader set of species including numerous isomeric forms. This work was focused on the validation of a UHPLC-MS/MS method for the simultaneous quantification of 21 sugars in PM, with a very simple and fast sample preparation. It presents the overall analytical method and its performances, including limits of quantification, linearity, intermediate precision studies, accuracy, and recovery. As a proof of concept, applicability, and to showcase challenging real-world samples, a timeseries of measurements of some species are presented for samples from Jungfraujoch (JFJ) site, Switzerland (3580 m a.s.l.), an alpine high-altitude international research station (Bukowiecki et al., 2016). Ultimately, the high sensitivity and ease of this method could pave the way for numerous additional studies on these families of chemical species, enhancing our understanding of atmospheric sugar cycles and their connection to biological processes.

### 2 Material and methods

#### 2.1 Standards and reagents







A large literature survey was conducted to make an assessment of the sugars, sugar alcohols and anhydrosugars that have previously been cited in the literature concerning atmospheric particulate matter, as synthetized in Table S1. A large selection of compounds was initially tested during the method development and 28 of them were kept for the development processes and method validation. Finally, only 21 of them were properly quantified and monitored for routine analysis. The chemical structures and physicochemical properties of these target compounds are shown in Table S2 and their suppliers are presented below.

Adonitol (BioXtra,  $\geq$  99.0% (HPLC), L-(+)-Arabinose (BioUltra,  $\geq$  99.5% (sum of enantiomers, HPLC)), L-(-)-Arabitol (≥ 98% (GC)), Erythritol (CRM), L-(+)-Erythrulose (≥ 85% HPLC), D-(-)-fructose (≥ 99% (HPLC)), D-(+)-galactose (≥ 99% (HPLC)), D-(+)-Glucose (BioUltra, anhydrous, ≥ 99.5% (sum of enantiomers, HPLC)), Glycerol (≥ 99.5%), Inositol (CRM), D-Lactose monohydrate (BioUltra, ≥ 99.5%), Levoglucosan (1,6-Anhydroβ-D-glucose (99%)), Maltitol (≥ 98%(HPLC)), D-(+)-Maltose monohydrate (BioUltra, ≥ 99.0%), D-Mannitol (BioUltra, ≥ 99.0% (sum of enantiomers, HPLC)), D-(+)-Mannose (synthetic, ≥ 99% (GC)), D-(+)-melezitose hydrate (≥ 97% (HPLC)), L-Rhamnose monohydrate (≥ 99%), D-(-)-Ribose (≥ 99% (GC)), Sedoheptulosan, Dsorbitol (BioUltra,  $\geq 99.0\%$  (HPLC)), Sucrose (Saccharose  $\geq 99.5\%$  (GC), BioXtra), D-Threitol (99%), D-(+)-Trehalose dihydrate (CRM), D-(+)-Xylose (≥ 99% (GC)) and D-(+)-Xylose (≥ 99% (GC)) were obtained from Merck-Sigma (France). Galactosan (1,6-Anhydro-beta-d-galactopyranose 97%) and Mannosan (1,6-Anhydrobeta-d-mannopyranose 97%) were purchased from Combi-Blocks (USA). 2-methyl-D-Erythritol (2S, 3R) and its 3 isomers (2-methyl-L-Erythritol (2R, 3S); 2-methyl-D-treitol (2S, 3S) and 2-methyl-L-treitol (2R, 3R)) were synthesized by Plateau Synthese Organique, Département de Chimie Moléculaire (University of Grenoble Alpes, France) according to (Ghosh et al., 2012). Isotopically labelled standards, myo-Inositol (1,2,3,4,5,6-D6, 98%), D-Glucose (1,2,3,4,5,6,6-D<sub>7</sub>, 97-98%) and 1,6-Anhydro-beta-d-glucose (Levoglucosan) (U-13C6,98%) were obtained from Eurisotop. Ammonia solution (25%, LC-MS grade), Water (Rotisolv®, LC-MS grade) and Acetonitrile (Rotisolv® ≥99.95%, LC-MS grade) were obtained from Roth.

## 2.2 Calibrators preparation

Individual stock solution of each target analyte and each internal standard are prepared in MilliQ® water (18.2 MΩ.cm) at 1000 ppm and stored at 4°C. A stock solution of all target analytes (MIX-28) is prepared by mixing each target analytes in MilliQ® water. In parallel, a stock solution of the three internal standards, inositol-D6, glucose-D7 and levoglucosan-13C6 (MIX-IS) is prepared in the same manner at 10 ppm. MIX-28 and MIX-IS water solutions are used to prepare an eight-point calibration curve in solvent with final concentrations of 90% acetonitrile, 0.005 % NH<sub>4</sub>OH and 8 ppb for internal standards. Calibration range for each target analyte was adapted from a wide variety of analysed real samples, and is displayed in Table 2.

## 2.3 UHPLC-MS/MS conditions

All measurements were performed with ultra-high-performance liquid chromatography (ExionLC – AD binary pump, Shimadzu) coupled to a tandem mass spectrometer (AB SCIEX 5500 QTRAP). The column oven was set at  $30^{\circ}$ C. The injection volume was  $30 \,\mu$ L. The chromatographic separation was performed on a Luna Omega Sugar (150 mm x 2.1 mm x 3  $\mu$ m) column from Phenomenex (France) with a compatible guard column (SecurityGuard

Cartridges, Sugar, 4 x 2.0 mm ID, Phenomenex). This column is of hydrophilic interaction chromatography (HILIC) type and is adapted to hydrophilic interaction and separation of hydrophilic compounds such as sugars, in aqueous matrix. One overall goal of this development was to get simplified water extraction of the samples, but it made direct injection impossible since HILIC mode supports only comparable sample and eluants initial conditions (10% water, 90% acetonitrile). Other columns with such specifications (BEH Amide 100 mm x 2.1 mm x 1.7  $\mu$ m from Waters, and HPLC Ultra Amino 150 mm x 3 mm x 3  $\mu$ m from Restek) were tested but did not provide such good results in terms of separation (peak shapes, return to baseline) after optimization, compared to the Luna Omega Sugar column.








Mobile phase (A) was 0.002% ammonia in H<sub>2</sub>O (LC-MS grade) and mobile phase (B) was pure acetonitrile (LC-MS grade). The gradient elution was programmed as follows (2 isocratic steps followed by the rinsing of the column and equilibration): 0.00 - 9.00 min, 10% mobile phase (A), 0.45 mL/min; 10.00 - 23.00 min, 20% mobile phase (A), 0.40 mL/min; 23.01 - 28.00 min, 40% mobile phase (A), 0.45 mL/min; 28.01 - 29.01 min, 10% mobile phase A, 0.45 mL/min; 29.01 - 36.00 min, 10% mobile phase A, 0.45 mL/min. Total run time was 36 min. In order to limit soiling of the MS system, the divert valve was programmed to MS analysis only between 1 and 25 min. The autosampler temperature was set to 5°C in order to limit potential evolution of the samples during the analytical batch.

The mass spectrometer was operated in negative electrospray ionization (ESI) mode using a scheduled multiple reaction monitoring (MRM). The ESI parameters were optimised and finally set to the following ones: spray voltage -4500 V, source temperature  $400^{\circ}$ C, Ion Source Gas 1 (GS1) 40.0 psi, Ion Source Gas 2 (GS2) 60.0 psi. Q1 and Q3 mass resolution is  $\pm 0.1$  Da. The MRM transition per compounds were determined by infusion of individual standards at 100 ppb in 10/90% water/acetonitrile, in presence of 0.005% NH4OH. For each target analyte, infusion of monospecific standard solution was made: a full scan mass spectrum was acquired to determine its precursor ion. Then, product ions scans were acquired on the selected precursor ion to select a product ion of interest. One MRM transition is chosen to characterize each compound. All MRM parameters for each analyte are summarized in Table 1.

Given the fact that some analytes are isomers, the identification of each analyte is also performed using its elution order and its retention time. Given the nature of the HILIC phase, retention time may slightly change between each batch, and a standard solution is used to recalibrate the retention time of each analyte before batch launches. The system is controlled with the Analyst® software version 1.7 (AB SCIEX). All data processing (integration peak and quantification) was conducted on SCIEX OS v1.7, using individual internal standards for each compound with the external calibration to correct potential ionisation variations in the source.

Table 1: Multiple Reaction Monitoring (MRM) parameters for all analytes and internal standards. DP: Declustering Potential; CE: Collision Energy; CXP: Collision Cell Exit Potential; \*: retention time can be adjusted according to the ageing of the column.

| Analyte                  | Q1<br>(m/z) | Q3<br>(m/z) | Retention time (min)* | MRM window (sec) | DP<br>(volts) | CE<br>(volts) | CXP<br>(volts) | IS   |
|--------------------------|-------------|-------------|-----------------------|------------------|---------------|---------------|----------------|------|
| Levoglucosan-13C6 (L-IS) | 166.9       | 105.0       | 2.38                  | 60.0             | -95           | -14           | -11            | -    |
| Glucose-D7 (G-IS)        | 186.0       | 60.9        | 11.21                 | 120.0            | -40           | -22           | -9             | -    |
| Inositol-D6 (I-IS)       | 184.9       | 89.0        | 15.20                 | 60.0             | -110          | -24           | -11            | -    |
| Adonitol                 | 150.9       | 89.0        | 5.75                  | 60.0             | -60           | -16           | -11            | L-IS |
| Arabinose                | 148.9       | 89.0        | 6.00                  | 120.0            | -30           | -10           | -11            | L-IS |

| Analyte          | Q1<br>(m/z) | Q3<br>(m/z) | Retention time (min)* | MRM window (sec) | DP<br>(volts) | CE<br>(volts) | CXP<br>(volts) | IS   |
|------------------|-------------|-------------|-----------------------|------------------|---------------|---------------|----------------|------|
| Arabitol         | 150.9       | 89.0        | 6.41                  | 120.0            | -75           | -16           | -11            | L-IS |
| Erythritol       | 120.9       | 89.0        | 3.83                  | 60.0             | -45           | -14           | -11            | L-IS |
| Erythrulose      | 119.1       | 70.9        | 2.03                  | 60.0             | -30           | -16           | -9             | L-IS |
| Fructose         | 178.9       | 89.0        | 7.78                  | 180.0            | -40           | -12           | -9             | G-IS |
| Galactosan       | 160.9       | 100.9       | 2.09                  | 60.0             | -75           | -16           | -11            | L-IS |
| Galactose        | 178.9       | 89.0        | 12.00                 | 90.0             | -65           | -12           | -11            | G-IS |
| Glucose          | 179.0       | 89.0        | 11.14                 | 200.0            | -55           | -10           | -11            | G-IS |
| Glycerol         | 90.9        | 59.0        | 2.41                  | 60.0             | -55           | -14           | -7             | L-IS |
| Inositol         | 178.9       | 86.9        | 15.07                 | 60.0             | -90           | -22           | -13            | I-IS |
| Lactose          | 341.0       | 161.0       | 17.85                 | 120.0            | -95           | -10           | -17            | I-IS |
| Levoglucosan     | 160.9       | 100.9       | 2.39                  | 60.0             | -70           | -14           | -11            | L-IS |
| Maltitol         | 343.0       | 179.0       | 16.55                 | 90.0             | -135          | -20           | -11            | I-IS |
| Maltose          | 341.0       | 161.0       | 16.29                 | 120.0            | -90           | -10           | -15            | I-IS |
| Mannitol         | 180.9       | 89.0        | 11.36                 | 120.0            | -100          | -18           | -11            | G-IS |
| Mannosan         | 160.9       | 100.9       | 2.09                  | 60.0             | -65           | -18           | -11            | L-IS |
| Mannose          | 178.9       | 89.0        | 10.25                 | 60.0             | -45           | -12           | -11            | G-IS |
| Melezitose       | 503.1       | 323.1       | 21.90                 | 90.0             | -180          | -28           | -15            | I-IS |
| Rhamnose         | 162.9       | 59.0        | 3.47                  | 60.0             | -60           | -18           | -9             | L-IS |
| Ribose           | 148.9       | 88.9        | 3.50                  | 90.0             | -45           | -10           | -11            | L-IS |
| Sedoheptulosan   | 190.9       | 116.9       | 6.66                  | 90.0             | -90           | -20           | -13            | L-IS |
| Sorbitol         | 180.9       | 88.9        | 10.71                 | 120.0            | -80           | -20           | -11            | G-IS |
| Sucrose          | 341.0       | 59.0        | 14.61                 | 60.0             | -130          | -54           | -7             | I-IS |
| Threitol         | 120.9       | 88.9        | 3.83                  | 120.0            | -55           | -14           | -11            | L-IS |
| Trehalose        | 341.0       | 58.9        | 17.48                 | 60.0             | -150          | -52           | -7             | I-IS |
| Xylose           | 148.9       | 59.0        | 4.67                  | 60.0             | -30           | -18           | -7             | L-IS |
| 2-methyl-tetrols | 134.9       | 85.0        | 2.76                  | 60.0             | -60           | -20           | -9             | L-IS |

# 2.4 Sample collection and preparation




Ambient aerosol samples were collected at the High Altitude Research Station Jungfraujoch (JFJ) (3580 m a.s.l., Switzerland) as part of the Swiss National Air Pollution Monitoring Network (NABEL). The JFJ station is located in the free troposphere with regular intrusion of planetary boundary layer air masses. It generally offers very low background aerosol concentrations. Samples were collected over 24h on quartz fiber filters (150 mm diameter discs, Pallflex 2500 QAT-UP) using DHA-80 high volume samplers (DIGITEL) with a PM10 inlet, operating at 45 m<sup>3</sup> h<sup>-1</sup>. After sampling, the filter samples are transported back to the laboratory for gravimetric determination of the PM10 mass concentration. Subsequently, the filters are put inside a glassine envelope (PAWI Packaging Schweiz AG), sealed in polyethylene bags and stored at -18°C until further analysis (Hueglin et al., 2005). The samples studied here were composites of 4 daily filters distributed over 2 weeks, for a period of several years from 2011-2016. A total of 136 composite samples and 11 field blanks were analysed.

For the extraction of analytes of interest, a portion of the quartz filter (typically a total of 10.2 cm<sup>2</sup> for each composite sample from the JFJ site) was used. The filter was extracted in 6 mL of MilliQ® water with vortex shaking for 20 min. This sample preparation was adjusted for JFJ samples for which very low concentrations of

sugars were expected. In cases of daily samples from most low altitude sites, 5 cm² of quartz filter are generally extracted in 7 mL of water (Aas et al., 2025). Then, the supernatant is filtered with 0.2 μm Ion Chromatography Acrodisc®13 pre-washed with ultrapure water. If necessary, extracts are stored in a freezer at -20°C for later analysis. Finally, 100 μL of the extract is diluted with 900 μL of the dilution mixture (99% acetonitrile, 0.005% NH<sub>4</sub>OH and internal standard adjusted at 9 ppb from MIX-IS) to reach chromatographic equilibration conditions, and then analysed by UHPLC-MS/MS. Samples stay 69 hours at most in the refrigerated autosampler before analysis, in order to avoid potential aging processes.

## 2.5 Method performance evaluation

#### 2.5.1 Method validation





The performance of the method was evaluated by studying linearity, limit of quantification (LOQ), limit of detection (LOD), precision, bias, and recovery. The linearity was evaluated with 8 points of calibration per compound. The calibration curves were determined using least-squares linear regressions. The LOD was determined as the minimal detected standard with a signal/noise ratio larger than 3. Precision studies including intra-day and inter-day evaluations, and bias were determined with the injection of a standard every 8 injections during each batch. Accuracy was evaluated for levoglucosan by comparison to standard reference material NIST® SRM® 2786 (Fine Atmospheric Particulate Matter). Recovery was evaluated on summer and winter filters collected on the roof of the laboratory in Grenoble (France), to test for a potential seasonal effect.

## 2.5.2 Calculation and analytical batch

The quantification is based on the isotope dilution with the addition of isotope labelled sugars (inositol-D6, glucose-D7 and levoglucosan-13C6) into the injected sample to correct analytical variation of the source of the mass spectrometer. Calibration curves are drawn by plotting the area ratio (Compound area/IS area) against the concentration ratio (Compound concentration/IS concentration). Thus, the sample concentration, in ppb, can be determined with the area ratio of the sample. Finally, results were converted in ng/m³ using the extracted surface of filter, the volume of air collected, the initial volume of water for the extraction, the dilution factor with the eluant, and taking into account the average of the sample field blanks of the corresponding field campaign. In daily-routine, full set of calibration solutions are injected every 40 samples (about every 24 hours of analysis), and the standard 3 of the calibration range (STD3) is injected every 8 injections as a quality control. These

repetitions are performed to ensure that the LC conditions, in particular column ageing, are compatible with an

### 235 3 Results and discussion

## 3.1 Sample preparation

accurate quantification of samples.

The sample preparation exclusively in water and without derivatization is intended to be simple, limiting the preparation stages, the handling, the possibilities of contamination, and the use of solvents and other chemicals. All of these convey major advantages for this preparation in terms of time consumption, consumables costs, and required manpower. Moreover, these same aqueous extracts can be directly used for other analyses such as major

ions and organic acids by ion chromatography coupled to mass spectrometer on anion canal (IC-MS) (Glojek et al., 2024), sugars analysis by IC-PAD (Samaké et al., 2019), or HUmic LIke Substances analysis using a TOC analyser after separation by HPLC-Fluorescent (Baduel et al., 2009). Therefore, these extracts, the associated costs and the consumption of the sample filter surface can be shared between these different analyses, which is another big advantage.

## 3.2 Method validation






#### 3.2.1 Retention times and co-elutions

The 28 compounds eluated along the 25.0 min MS analysis period (Figure 1) and retention times were repeatable with a relative standard deviation (RSD) below 2.0% for all compounds for 146 successive injections over a period of 61 hours. However, some chemical species cannot be separated in this run.

First, the 2-methyl-erythritol and 2-methyl-threitol four stereoisomers are co-eluted and present identical MRM transition, but they also provide the same response factor. 2-methyl-D-Erythritol is then used as the quantification standard and represents therefore the four isomers, named commonly as 2-methyl-tetrols. Second, the two compounds in each pair galactosan/mannosan and erythritol/threitol present the same retention time and the same MRM transitions, but with different response factors in terms of intensity for identical concentrations. Their quantification cannot be achieved with this method.

Finally, the Luna Omega Sugar column was globally suitable for the analysis of the other sugar compounds from Table 1, but it must be replaced around every 300-600 injections due to the column ageing. Indeed, a drift of retention time was observed with ageing, with (for the example of figure S1) an average over all 28 compounds of 0.5 min after 512 injections, which can go up to 2.2 min for the case of melezitose at the end of the run (Figure S1). This ageing may be due to anion (such as Cl<sup>-</sup> from extract samples) progressive occupation of active separation sites (like amine), leading to less effective separation of sugars and retention time decrease through numerous and charged sample injections (personal communication J. Lacouchie-Payen, Phenomenex). The pairs adonitol/arabitol and sorbitol/mannitol tend to co-eluate with increased column ageing (Figure S1). This cannot be reversed with any cleaning procedure. A criterion to discard an aged column is the difficulty to separate completely those 2 pairs of compounds.

Figure 1: Chromatogram of standard 1 (STD1), the highest concentrated standard of calibration range, showing the elution of the 28 sugars, sugar alcohols and anhydro-sugars on the Luna Omega Sugar column.

#### 3.2.2 Cases of tested but not analyzable compounds






Several compounds were tested and monitored but no result can be returned. This is the case for arabinose and galactose which were not automatically detected in most of the analytical batches due to their weak intensity and their elution at the end of the trailing peak of xylose and glucose, respectively. Therefore, linearity cannot be evaluated, neither proper quantification. However, our experience with many series of samples in various atmospheric conditions is that they are not present in the atmospheric PM10 with the LOD mentioned in Table 2. Finally, glycerol is also a specific case. It has been reported in plants in high concentrations (Gerber et al., 1988) and has an influence on plant flowering (Lazare et al., 2019). Moreover, Kang et al. (2017) found higher level of glycerol in PM2.5 filter samples during winter and autumn when vegetation decays and fungal population increases. Several studies worldwide confirmed the presence of glycerol in PM and associated it with biomass burning (Zangrando et al., 2016). Our results of the method performance evaluation showed that glycerol can be easily contaminated. This can take place at any time from the sampling to the LC-MS/MS analysis, as currently glycerol is present in everyday life (Bagnato et al., 2017) as it is included in a large variety of products such as cosmetics, food (E422, (EFSA Panel on Food Additives and Flavourings (FAF) et al., 2022), pharmaceutical and tobacco products in particular e-cigarettes (Kubica, 2023). We choose not to report glycerol concentrations since results obtained are often (but not always) erratic. More work must be done for understanding the contamination processes, limiting them, and obtain trustable results.

### 3.2.3 Linearity and Limit of Detection (LOD)

The calibration range defined for this method was determined for each compound in order to represent the full range of the concentrations generally expected from atmospheric samples prepared with our usual sampling and extraction conditions. This was determined following our experience for the compounds measured from a large literature review. The linearity was evaluated for the full 8 points calibration range of each compound. A very

good correlation between prescribed and observed concentrations were observed with r<sup>2</sup> above 0.99 (Pearson's criteria), except for maltitol and sorbitol (Table 2).

The analytical Limit of Detection (LOD) is the smallest standard detected in the analytical run, with a signal/noise ratio > 3. Values are displayed in ppb in the extract. The analytical LOD is compound-dependent and varies from 0.001 - 2.5 ppb, and after conversion in ng/m³ using our standard sampling and extraction conditions, from 0.001 - 3.6 ng/m³ (Table 2). However, the actual field LOD will vary according to the field blank value, for each compound and each field campaign.

## 3.2.4 Intermediate precision studies




In our classical analytical batch, the standard 3 (STD3) is injected every 8 injections as a quality control. Bias is calculated for STD3 as follows (Eq. 1):

Bias = 
$$100 - \frac{Experimental concentration of STD3}{Theoretical concentration of STD3} \times 100$$
 (1)

94% of intra- and inter-day bias results obtained over a period of 5 days were below a 25% bias, meaning that the method quantification is robust (Table 2). 83% of intra- and inter-day RSD results obtained over a period of 5 days were also below 25%, meaning that this UHPLC-MS/MS method is repeatable and reproducible (Table 2). We note that for intra-day, more RSD results (n=6) were larger than 25%, compared to inter-day RSD results (n=3). However, for the compounds with particular interest (like the ones used in source apportionment studies such as arabitol, fructose, glucose, inositol, levoglucosan, mannitol, sucrose, trehalose and 2-methyl-tetrols), RSD results were similar between intra-, and inter- day. Conversely, the species with the worst results happen to be of lower interest for atmospheric samples, with maltitol never being detected with a LOD of 0.05 ng/m³, and sorbitol rarely seen with a LOD of 0.005 ng/m³.

Table 2 : Performance of the sugars UHPLC-MS/MS method. RSD: Relative Standard Deviation; LOD: Limit of Detection; STD3: standard 3 of the calibration range.

| Compounds    | Calibration Linearity | STD3 Concentration – | Intra-day<br>(n=13) |             | Inter-day<br>(n=5) |             | Median<br>LOD | Median<br>LOD |                  |
|--------------|-----------------------|----------------------|---------------------|-------------|--------------------|-------------|---------------|---------------|------------------|
|              | range (ppb)           | (r²)                 | (ppb)               | Bias<br>(%) | RSD<br>(%)         | Bias<br>(%) | RSD<br>(%)    |               | (ng/m3)<br>(n=5) |
| Adonitol     | 0.001 - 10            | 0.9992               | 1.000               | -7.6        | 17.3               | -6.7        | 16.6          | 0.001         | 0.011            |
| Arabinose    | 0.005 - 50            | N/A                  | 4.993               | N/A         | N/A                | N/A         | N/A           | 2.497         | 3.614            |
| Arabitol     | 0.010 -100            | 0.9997               | 10.000              | -8.9        | 21.9               | -11.0       | 18.5          | 0.010         | 0.011            |
| Erythrulose  | 0.001 - 10            | 0.9985               | 1.001               | -10.2       | 16.6               | -0.5        | 17.2          | 0.010         | 0.108            |
| Fructose     | 0.001 -10             | 0.9959               | 0.994               | 38.8        | 15.2               | -10.6       | 15.5          | 0.001         | 0.001            |
| Galactose    | 0.001 -10             | 0.8904               | 1.000               | N/A         | N/A                | N/A         | N/A           | 0.010         | 0.050            |
| Glucose      | 0.010 -100            | 0.9995               | 9.938               | 3.5         | 3.2                | -1.7        | 7.9           | 0.010         | 0.042            |
| Glycerol     | 0.050 -500            | 0.9986               | 50.000              | 17.7        | 9.4                | 12.9        | 17.5          | 0.050         | 0.054            |
| Inositol     | 0.001 -10             | 0.9995               | 1.010               | -0.9        | 6.2                | -4.1        | 5.7           | 0.051         | 0.055            |
| Lactose      | 0.001 -10             | 0.9998               | 0.998               | -8.0        | 20.0               | -9.2        | 17.0          | 0.050         | 0.054            |
| Levoglucosan | 0.100 -1000           | 0.9998               | 100.380             | 3.6         | 5.8                | 1.3         | 4.1           | 0.100         | 1.084            |
| Maltitol     | 0.001 -10             | 0.9684               | 1.004               | -49.7       | 57.4               | -23.9       | 23.2          | 0.050         | 0.054            |
| Maltose      | 0.001 -10             | 0.9981               | 0.996               | -7.5        | 17.0               | -6.2        | 18.2          | 0.050         | 0.042            |

| Commonada        | Calibration  | Linearity         | STD3                     | Intra-day<br>(n=13) |            | Inter-day<br>(n=5) |            | Median<br>LOD  | Median<br>LOD    |
|------------------|--------------|-------------------|--------------------------|---------------------|------------|--------------------|------------|----------------|------------------|
| Compounds        | range (ppb)  | (r <sup>2</sup> ) | Concentration -<br>(ppb) | Bias<br>(%)         | RSD<br>(%) | Bias<br>(%)        | RSD<br>(%) | (ppb)<br>(n=5) | (ng/m3)<br>(n=5) |
| Mannitol         | 0.050 - 500  | 0.9988            | 49.600                   | -12.5               | 26.5       | -19.2              | 21.5       | 0.050          | 0.054            |
| Mannose          | 0.010 - 100  | 0.9991            | 10.000                   | 2.1                 | 26.9       | -1.4               | 27.8       | 0.100          | 0.540            |
| Melezitose       | 0.001 -10    | 0.9989            | 1.006                    | 7.6                 | 45.4       | -7.8               | 28.5       | 0.010          | 0.011            |
| Rhamnose         | 0.010 - 100  | 0.9996            | 10.060                   | -7.6                | 5.6        | -3.9               | 10.0       | 0.010          | 0.109            |
| Ribose           | 0.005 -50    | 0.9936            | 5.000                    | -12.1               | 28.5       | -15.5              | 21.4       | 0.250          | 0.270            |
| Sedoheptulosan   | 0.001 -10    | 0.9991            | 1.008                    | 3.9                 | 6.0        | -2.5               | 10.8       | 0.050          | 0.109            |
| Sorbitol         | 0.005 - 50   | 0.9466            | 4.959                    | -63.7               | 110.8      | -14.2              | 54.5       | 0.005          | 0.005            |
| Sucrose          | 0.005 -50    | 0.9941            | 5.040                    | 19.5                | 10.0       | 14.5               | 9.9        | 0.005          | 0.005            |
| Trehalose        | 0.010 - 100  | 0.9996            | 9.920                    | 8.0                 | 9.9        | -0.6               | 8.4        | 0.100          | 0.107            |
| Xylose           | 0.100 - 1000 | 0.9996            | 99.800                   | -10.2               | 8.0        | -5.3               | 12.3       | 0.100          | 0.108            |
| 2-methyl-tetrols | 0.005 - 50   | 0.9997            | 4.494                    | 7.4                 | 7.5        | 3.5                | 6.4        | 0.004          | 0.007            |

# 3.2.5 Accuracy





Accuracy was evaluated for levoglucosan by aqueous extraction and UHPLC-MS/MS analysis of fine particulate matter reference material of Standard Reference Material (SRM®) 2786 from the National Institute of Standards and Technology (NIST®) (NIST® SRM® 2786) (n=6). Precision was 4.2% and bias was 10.0% which were compliant with RSD <15% and bias ±15%. In this SRM, concentrations for galactosan and mannosan are also available, but due to coelution and identical MRM transitions with different response factors in terms of intensity for identical concentration, their quantification is not guaranteed with this method. Currently, to our knowledge, no SRM is available for the other sugars monitored in this UHPLC-MS/MS method. However, for information purposes only, results of the concentrations (mg/kg) for 21 sugars of the NIST® SRM® 2786 are provided in Table S3.

# 3.2.6 Recovery

Extraction yield was assessed by comparing results from samples spiked at the STD3 level before and after extraction. The experiment was performed on summer (n=3) and winter filters (n=3) collected in the urban background atmosphere of Grenoble. The average chemical composition of the PM10 for a large range of compounds is described in Borlaza et al., (2021). It should be noted that several of the compounds presented in Table 3 present very low concentrations (sub-ng/m³ or lower), explaining some of the very high results (like sedoheptulosan). However, mean extraction yields were satisfying with most of the results between 70 and 120% (Table 3).

Table 3: Extraction yield evaluated on summer and winter filters.

|           | Extraction yield (%) |                     |      |  |  |  |
|-----------|----------------------|---------------------|------|--|--|--|
| Compounds | Summer filter (n=3)  | Winter filter (n=3) | Mean |  |  |  |
| Adonitol  | 82.4                 | 88.2                | 85.3 |  |  |  |
| Arabitol  | 83.7                 | 89.3                | 86.5 |  |  |  |

|                  | Ex                  | Extraction yield (%) |       |  |  |  |  |
|------------------|---------------------|----------------------|-------|--|--|--|--|
| Compounds        | Summer filter (n=3) | Winter filter (n=3)  | Mean  |  |  |  |  |
| Erythrulose      | N/A                 | N/A                  | N/A   |  |  |  |  |
| Fructose         | 83.9                | 90.8                 | 87.4  |  |  |  |  |
| Glucose          | 97.8                | 122                  | 110   |  |  |  |  |
| Glycerol         | 80.5                | 90.3                 | 85.4  |  |  |  |  |
| Inositol         | 81.1                | 88.1                 | 84.6  |  |  |  |  |
| Lactose          | 84.0                | 84.2                 | 84.1  |  |  |  |  |
| Levoglucosan     | 77.2                | 94.5                 | 85.8  |  |  |  |  |
| Maltitol         | 53.9                | 71.5                 | 62.7  |  |  |  |  |
| Maltose          | 91.9                | 68.5                 | 80.2  |  |  |  |  |
| Mannitol         | 77.0                | 84.4                 | 80.7  |  |  |  |  |
| Mannose          | 81.4                | 87.6                 | 84.5  |  |  |  |  |
| Melezitose       | 88.0                | 101                  | 94.6  |  |  |  |  |
| Rhamnose         | 81.5                | 97.9                 | 89.7  |  |  |  |  |
| Sedoheptulosan   | 702                 | 93.0                 | 82.2  |  |  |  |  |
| Sorbitol         | 57.6                | 78.8                 | 68.2  |  |  |  |  |
| Sucrose          | 201                 | 110                  | 155.4 |  |  |  |  |
| Trehalose        | 80.7                | 100                  | 90.4  |  |  |  |  |
| Xylose           | 79.5                | 81.0                 | 80.3  |  |  |  |  |
| 2-methyl-tetrols | 89.0                | 90.2                 | 89.6  |  |  |  |  |

In summary, this UHPLC-MS/MS method provides reliable and low-level quantification for 21 sugars for atmospheric fine particles including sugars (erythrulose, fructose, glucose, lactose, maltose, mannose, melezitose, rhamnose, ribose, sucrose, trehalose, xylose), sugar alcohols (adonitol, arabitol, inositol, maltitol, mannitol, sorbitol, 2-methyl-tetrols) and anhydro-sugars (levoglucosan, sedoheptulosan). New sugars (adonitol, erythrulose, fructose, maltitol, maltose, melezitose, sedoheptulosan, sucrose, trehalose, xylose and 2-methyl-tetrols) not quantified with traditional IC-PAD method (Engling et al., 2006; Yttri et al., 2007; Samaké et al., 2019) are provided and will allow further exploration of their sources and the processes they undergo in the atmosphere.

# 3.3 Jungfraujoch 6-year time series of samples

# 345 3.3.1 Atmospheric concentration calculations



The interest of this UHPLC-MS/MS method is demonstrated with the analysis of a long time series of samples collected in the low concentration levels environment of the High Altitude Research Station Jungfraujoch (Switzerland). These samples can be regarded as challenging, insofar as they usually contain very low PM values and hence sugar concentrations. With the simple and fast sample preparation method, the 136 composite samples and 11 fields blanks were prepared in two days. After analysis, the results of composites and field blanks were processed in the same manner in SCIEX OS. The field limit of quantification (Field LOQ) was calculated for each compound (Table S4), based on the field blanks results, as follows: (Eq. 2)

Field LOQ = Mean field blanks 
$$+ 2 x$$
 Standard Deviation field blanks

When results for composite samples were above this Field LOQ, the value of mean field blank was subtracted from the initial sample value in ppb, subsequently converted in ng/m<sup>3</sup>.

## 3.3.2 Evolution of the concentrations







Median and seasonal concentrations of the 21 quantifiable sugars are presented in Table S4. The time series of 3 sugars of interest (arabitol, 2-methyl-tetrols and levoglucosan) are plotted on Figure 2 over a 6-year period, with a time resolution of 20 composite samples per year.

Arabitol displays values between LOD (0.011 ng/m³) and 2 ng/m³, with seasonal variation patterns (May-November occurrence) and maximal values in high summer. Interannual variability of the maxima summer concentrations are between 0.6 and 2 ng/m³. Following the same pattern as arabitol, the concentrations of other primary biogenic (PBOA) sugars such as mannitol, glucose, fructose, trehalose, and melezitose also display strong seasonal variations with much larger concentration in summer that are below LOD in winter. All concentrations (Table S4) are much lower (by a factor of about 20) than usual concentrations in low-altitude environments in France, and also in Alpine valleys (e.g. in Grenoble (Samaké et al., 2019) or Slovenian valleys (Glojek et al., 2024)).

In comparison, the secondary organic aerosol (SOA) tracer 2-methyl-tetrols, associated with the oxidation products from isoprene emissions, also displays strong seasonality with inter-annual variability of concentrations, but with a shorter occurrence of this compound compared to arabitol, restricted to June-to-early September (Figure 2). Contrary to primary biogenic sugars, concentrations reach much higher values (between 5 and 25 ng/m³ for each summer maximum), and are only 4 to 12 times lower than in Alpine lower altitudes (Glojek et al., 2024).

concentrations are low (

Figure 2: Evolution of sugars arabitol, 2-methyl-tetrols and levoglucosan concentrations in the air (ng/m³) over years 2011-2016 at Jungfraujoch.

### 4 Conclusion




This UHPLC-MS/MS method for molecular-resolution quantification of sugars in atmospheric PM10 samples offers several advantages. First, the sample preparation is intended to be simple, with an extraction in water, without any other steps like solvent reduction or derivatization. This conveys a major advantage of this preparation in terms of time consumption, consumables costs and required manpower. The preparation of the series of samples (n = 147) taken as an example in this work was performed in only two days. Moreover, this sample preparation can be adjusted according to the type of samples, including low concentrations as in this work.

Analytical results were very good in terms of linearity (r<sup>2</sup>>0.99), precision (RSD<25%), extraction yields (results between 70 and 120%). Accuracy was compliant by comparison with the NIST® SRM® 2786. This method offers

a larger number of sugar compounds properly quantified than with traditional IC-PAD method), which will allow enhanced exploration of sources and fate of a larger array of sugars in the atmosphere.

This highly sensitive method was successfully applied to 136 composite samples and 11 field blanks from Jungfraujoch (JFJ), Switzerland, a high latitude alpine station. These JFJ results provide concentrations for the European free troposphere, and indicate large seasonal variations over a 6-year period for many compounds, including arabitol, 2-methyl-tetrols and levoglucosan.

The high-resolution molecular-level chemical speciation achieved with this approach enables to further explore sugars sources and processes. This method has already proven effective across a range of sites and sample types: PM10 samples in Alpine valley sites, like in Kanal (Slovenia) (Glojek et al., 2024) and Grenoble (France) (Cruaud et al., in prep), but also in cloud water (Bianco et al., 2025), snow samples (Schivalocchi et al., in prep) and ice cores samples (Piot et al., in prep). The wide applicability across matrix—from atmospheric aerosols to snow, cloud water, and ice cores—demonstrates its potential to greatly enhance our understanding of the dynamics of sugar compounds in various environmental compartments. As such, this method opens new opportunities for indepth investigations into the sources, transformations, and transport of sugars in the atmosphere, as well as their interactions with climatic and biological processes.

#### Author contributions



PB: Investigation, Writing - Original Draft, Visualization, Writing - Review & Editing; SD: Methodology, Investigation, Writing - Review & Editing, Visualization, Supervision; VJ: Methodology; RE: Resources, Data
 Curation; DT: Investigation; TR: Investigation; JW: Investigation, Writing - Review & Editing; PW: Resources, Writing - Review & Editing, Funding acquisition; IEH: Resources, Writing - Review & Editing, Funding acquisition; GU: Funding acquisition; J-LJ: Conceptualisation, Writing - Review & Editing, Supervision, Funding acquisition, Project administration

## Competing interests

The authors declare that they have no competing interests.

# Acknowledgments

We are thankful to Pierre-Yves Chavant and Mathieu Curtil from Plateau Synthèse Organique at Département de Chimie Moléculaire of the University of Grenoble Alpes (France) for the synthesis of 2methyl-D-Erythritol and its isomers.

Sample collection was carried out by the Swiss national air pollution monitoring network, NABEL. Sample preparation and the chemical analysis were performed on the AirOSol platform (IGE, Grenoble, France).

# Financial support

The Sciex 5500 QTRAP LC-MS/MS was funded by a grant from Labex OSUG@2020 (investissements d'avenir - ANR10-LABX56).

The ANR program Atmospheric Biogenic Sugar ANR-21-CE01-0021-01 provided the financial support for the analytical development, method validation and part of this collaboration on Jungfraujoch atmospheric samples. Part of the Jungfraujoch analysis was funded by the Swiss National Science Foundation (SNSF grant number: 202178), SNSF Science Booster crowdfunding (support by individuals, as well as Digitel AG and Camfil AG).

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
