# Peer review of "Quantification of 21 sugars in tropospheric particulate matter by ultra-high-performance liquid chromatography tandem mass spectrometry"

_EGUsphere, 2025_

## Referee Comment (RC1)

**Report to the manuscript „Quantification of 21 sugars in tropospheric particulate matter by ultra-high-performance liquid chromatography tandem mass spectrometry"**

This manuscript presents a complete method to analyze sugars from particulate matter samples with ultra-high-performance liquid-chromatography tandem mass spectrometry. The authors made an in-depth validation of the overall method in terms of common analytical criteria and provide precise instructions including its limitations. The method is also exemplarily applied to several real samples from a station at the "Jungfraujoch", spanning a time series of five years. Clearly observed are seasonal trends for three of the investigated sugars.

In accordance with AMTs referee guideline, following aspects are addressed:

1. Does the paper address relevant scientific questions within the scope of AMT?

--- Yes. For scientists in this field this manuscript provides insightful and well evaluated details on how to setup a sensitive HPLC-MS/MS method to analyze sugars from particulate matter.

2. Does the paper present novel concepts, ideas, tools, or data?

--- From my perspective the analysis of sugars with LC-MS is not new, many papers and thesis in this regard are out there dealing with that topic, in particular in the field of food and plant sciences. In fact, I wonder why the authors do not mention papers from these fields. Of course, the sample treatment is different, but the LC-MS methods should be quite comparable, if I am not mistaken. Therefore, I suggest that the authors provide and discuss some references also to these fields.

3. Are substantial conclusions reached?

--- Despite the insightful LC-MS development on that specific instruments no further substantial conclusions are reached.

4. Are the scientific methods and assumptions valid and clearly outlined?

--- The applied analytical method is sound and clearly outlined.

5. Are the results sufficient to support the interpretations and conclusions?

--- Not applicable, since no major conclusions are drawn. It is more a presentation of the validity of the method itself.

6. Is the description of experiments and calculations sufficiently complete and precise to allow their reproduction by fellow scientists (traceability of results)?

--- Yes.

7. Do the authors give proper credit to related work and clearly indicate their own new/original contribution?

--- As pointed out in 2., the authors should probably check on publications in the field of food and plant sciences for LC-MS methods on sugars and also reference those.

8. Does the title clearly reflect the contents of the paper?

--- Yes.

9. Does the abstract provide a concise and complete summary?

--- Yes.

10. Is the overall presentation well-structured and clear?

--- Appropriate.

11. Is the language fluent and precise?

--- Yes.

12. Are mathematical formulae, symbols, abbreviations, and units correctly defined and used?

--- Yes.

13. Should any parts of the paper (text, formulae, figures, tables) be clarified, reduced, combined, or eliminated?

--- In table 1, the number of decimals for the Q1 and Q3 settings should be reduced to one, so instead of 166.906 please write 166.9. The quadrupoles of that instrument are not able to resolve the m/z by the third digit, so providing these in the list of settings might be misleading to as providing data of a high-resolution mass spectrometer.

14. Are the number and quality of references appropriate?

--- As stated in point 2. and 7., the authors should probably check on publications in the field of food and plant sciences for LC-MS methods on sugars and also reference those.

15. Is the amount and quality of supplementary material appropriate?

--- Yes.

**In compliance with the AMT referee guideline I do recommend publishing this article, however, with some minor changes/additions and requested comments presented in the following:**

**(i)** Please, also look at publications from the field of food and plant sciences dealing with the analysis of sugars by LC-MS. As far as I am aware, there are several available, which in principal use similar MS methods and should thus be referenced and compared to your own method.

**(ii)** In table 1, the number of decimals for the Q1 and Q3 settings should be reduced to one, so, e.g., instead of 166.906 please write 166.9. The instrument used is not a high-resolution mass spectrometer.

---

## Author Comment (AC1)

**EGUsphere-2025-1951 by Bros, P., Darfeuil S., et al. 2025**

**Responses to reviewers**

We appreciate the reviewers' attention and suggestions for analysing our manuscript, and the generally very positive appraisal of our paper by both of them. The answers to the points raised by the two reviewers are detailed below.

**Reviewer#1:**

**(1) Please, also look at publications from the field of food and plant sciences dealing with the analysis of sugars by LC-MS. As far as I am aware, there are several available, which in principal use similar MS methods and should thus be referenced and compared to your own method.**

**Reply :** We agree with the reviewer and we added a paragraph in the introduction about sugars analysis by LC-MS/MS in plants, food and forensic domains.

***New paragraph :*** *(Lines 92-108) The analysis of sugars by LC-MS/MS is not a new topic, as they have been quantified for several years in different fields including food, plant and forensic fields. Indeed, fructose, glucose and sucrose are widely analysed in plant and food using such methods: in plants such as arabidopsis and rice (Ito et al., 2014), potatoes and strawberry (Georgelis et al., 2018), grapes (Gika et al., 2012), in bread (Nielsen et al., 2006), fruit juices (Rego et al., 2018), honeydew (Nguyen et al., 2020) and fermentation processes (Pismennõi et al., 2021). In a completely different field, a LC-MS/MS method was validated to analyse eleven sugars and sugar alcohols related to explosive (Tsai et al., 2022). The most comprehensive methods reported to date allow the simultaneous analysis of no more than 19 sugar compounds (Gika et al., 2012; Ito et al., 2014). Even if application areas were numerous, the implemented LC-MS methods were essentially the same. Indeed, the LC separation was performed in an HILIC (Hydrophilic Interaction Liquid Chromatography) mode with column coated with amine or polyamide groups in order to enhance the polar compound retention and selectivity. Moreover, the MS was operated in negative mode and mostly using multiple reaction monitoring (MRM). In summary, these runs exhibit certain similarities with our intended approach, but remain constrained with respect to the range of analytes quantified and are most likely restricted to substantially higher concentration levels than expected for atmospheric PM samples, though probably impacted by more complex matrices. These observations collectively indicate that our experimental design is feasible; however, its implementation require substantial methodological refinement and further development to extend the analytical scope, particularly to encompass a broader set of species including numerous isomeric forms.*

Associated new references are highlighted in red in the "References" section.

**(2) In table 1, the number of decimals for the Q1 and Q3 settings should be reduced to one, so, e.g., instead of 166.906 please write 166.9. The instrument used is not a high-resolution mass spectrometer.**

**Reply :** We agree with the reviewer and reduce the number of digits for Q1 and Q3 as the QTRAP is a low resolution mass spectrometer. Table 1 has been modified accordingly and a sentence was added on the resolution.

*New sentence: (line 176) Q1 and Q3 mass resolution is ±0.1 Da.*

**(1) Add the mass resolution of the mass spectrometer. In Table 1 m/z is reported in 3 digits. However, isomers show different m/z, although they should be the same (e.g. fructose, glucose). Probably mass resolution is not that high.**

**Reply :** We agree with the reviewer. We reduce the number of digits for Q1 and Q" to 0.1. Indeed, a Triple Quadrupole is a low-resolution mass spectrometer with resolution around 0.1 Da. Table 1 has been modified accordingly and a sentence was added on the resolution.

*New sentence: (line 176) Q1 and Q3 mass resolution is ±0.1 Da.*

**(2) Equation 1and Table 2: Why do you use an absolute value for bias? Thus, the direction of the bias cannot be seen.**

**Reply :** We agree with the reviewer, the direction of the bias cannot be seen by using absolute value for bias. Equation 1 and Table 2 were modified accordingly.

**(3) Table S2: You do not report the molecular mass but Q1. Clarify this.**

**Reply :** Thank you for the comment, it was a mistake. Values in Table S2 were modified to display molecular mass and not Q1 values.

**(4) Line 66; Sentence not clear (However…)**

**Reply : The sentence was modified.**

*New sentence : (Lines 66- 69) However, this fungi-sugar link has only been documented for a limited number of sugars (e.g. arabitol, mannitol) and cases (Marynowski et al., 2020; Samaké et al., 2020), and the main sources and drivers of many other S and SA in PM have not been identified yet.*

**(5) Line 80: replace tgenerally by generally**

**Reply :** Thank you for the comment, "tgenerally" was replaced by "is generally" (line 80, in red).

**(6) Line 120: replace "et" by and**

**Reply :** Thank you for the comment, "et" was replaced by "and" (line 138, in red).

**(7) Line 128: not correct units of MQ-water**

**Reply :** Thank you for the comment, the unit of MilliQ-water was modified into "MΩ.cm" (line 146, in red).

**(8) Line 130: MIX-SI stands for mixture of internal standards. I recommend using MIX-IS. I assume you used the French abbreviation.**

**Reply :** We agree with the reviewer and replace "MIX-SI" by "MIS-IS" in lines 148 and 210 (in red).

**(9) Line 305: others, replace by other**

**Reply :** Thank you for the comment, "others" was replaced by "other" in line 324 (in red).